# Communicative competence assessment for learning: The effect of the application of a model on teachers in Spain

**Juan Jesús Torres-Gordillo**[1]*, **Fernando Guzmán-Simón**[2], **Beatriz García-Ortiz**[1]

1 Department of Educational Research Methods and Diagnostics, Educational Sciences Faculty, University of Seville, Seville, Spain, 2 Department of Language and Literature Teaching, Educational Sciences Faculty, University of Seville, Seville, Spain

* juanj@us.es

## Abstract

The evolution of the results of Progress in International Reading Literacy Study in 2006, 2011 and 2016, as well as the difficulties found by teachers implementing the core competences, have led to the need to reflect on new assessment models. The objective of our research was to design a communicative competence assessment model and verify its effect on primary education teachers. The method applied was a focus group study. Participants came from four primary education schools in the province of Seville (Spain). The data were gathered through discussion groups. The COREQ checklist was followed. Qualitative thematic analysis of the data was carried out using Atlas-ti. An inductive coding scheme was established. The results have enabled the construction of a communicative competence assessment model and its application in primary education classrooms with HERACLES. The effects of the assessment model and the computer software were different according to teachers' profiles. On the one hand, teachers open to educational innovation remained positive when facing the systematic and thorough assessment model. On the other hand, teachers less receptive to changes considered the model to be complex and difficult to apply in the classroom. In conclusion, HERACLES had a beneficial effect on communicative competence assessment throughout the curriculum and made teachers aware of the different dimensions of communicative competence (speaking, listening, reading and writing) and discourse levels (genre, macrostructure and microstructure).

**Data Availability Statement:** All relevant data are within the paper and its Supporting Information files.

## 1 Introduction

Assessments carried out by the International Association for the Evaluation of Educational Achievement (IEA) in Spain have provided new evidence for the effects of the educational improvement measures applied in primary education in the last two decades. In particular, the assessment performed in Progress in International Reading Literacy Study (PIRLS) in 2006, 2011 and 2016 has shown how competence in communication in Spanish primary education has not progressed at the rate of that of other European countries [1–3].

**Funding:** This work was supported by the Ministry of Economics and Competitiveness of Spain [grant number EDU2013-44176-P] as an R+D project entitled "Mejora de la Competencia en Comunicación Lingüística del alumnado de Educación Infantil y Educación Primaria" ("Improvement of the Competence in Linguistic Communication of Early Childhood Education and Primary Education", obtained in a competitive call corresponding to the State Plan of Advancement of Scientific and Technical Research of Excellence, State Subprogram of Generation of Knowledge, of the Secretary of State for Research, Development and Innovation.

**Competing interests:** The authors have declared that no competing interests exist.

The Spanish government and different regional authorities have implemented diverse improvement plans, which have been focused on the modifications of the official curriculum and on educational legislation to reverse this situation [4–6]. However, their results have not been expected in the area of communicative competence. Today, we are familiar with numerous definitions of communicative competence [7–13]. The publication in 2001 of the *Common European Framework of Reference for Languages* [14] has enabled us to describe the skills required for communication and their levels of achievement related to reading, writing, listening and speaking.

Moreover, the development of communicative competence in the educational curriculum must be related to 'accountability' within teaching programmes, which leads us to delve deeper into the link had by the school curriculum, based on key competences and their assessment. Training in the assessment of competences in general and of communicative competence in particular presents numerous deficiencies in the initial and continuous training of primary education teachers. Similarly, the difficulty in adapting the communicative competence theoretical concept to assessment in classrooms has led numerous authors to analyse the need to incorporate linguistic, cultural and social elements into the current educational context [15, 16]. Consequently, our paper focuses on the design and evaluation of a model for the assessment of communicative competence based on the Spanish curriculum through the use of a custom-designed computer application.

## 1.1 Assessment for learning

The design of an assessment model of communicative competence in the school context requires prior reflection related to the dimension assessment model, first, and an assessment of communicative competence, second. Our research started with a reflection on which assessment model for learning was the most appropriate for incorporating communicative competence assessment in the primary education classroom. Assessment for learning is considered an assessment that fosters students' learning [17–19]. Wiliam and Thompson [20] have developed five key strategies that enable this process to become an educational assessment:

1. Clarify and share the learning intentions and criteria for success.

2. Conduct effective discussions in the classroom and other learning tasks, which provide us with evidence of students' comprehension.

3. Provide feedback, which allows students to progress.

4. Activate the students themselves as didactic resources for their peers.

5. Foster the students as masters of their own learning.

The assessment that truly supports learning has two characteristics [21]: the feedback generated must provide information about learning activities for the improvement of performance, and the student must participate in actions for the improvement of learning based on heteroassessment, peer assessment and self-assessment.

Assessment for learning must set out by gathering information, which enables teachers and learners to be able to use it for feedback; that is, the result of the assessment must be information that both the teacher and the student can interpret for the improvement of the task. Wiliam [21] proposes an assessment that is incorporated into classroom programming and the information of which is relevant for the improvement of the teaching-learning process. Decision making for the improvement of the task must be based on the information that the

assessment indicators contribute to the learning process. In conclusion, the effort that the school makes to emphasise the learning assessment is justified for the following reasons:

1. Assessment must not be limited to marking (summative assessment); rather, it has to do with helping students learn [22].

2. Assessment is a key element in effective teaching, as it measures the results of learning addressed in the teaching-learning process [21].

3. Feedback plays a fundamental role and requires the information received to be used by students to improve their learning [23].

4. Instead of being content with solving obstacles in students' learning, teachers must offer opportunities from the assessment to develop learning strategies [24].

## 1.2 Communicative competence in the European educational framework

The theoretical construct on which our research is based has different sources. Since the 1960s, *communicative competence* has been approached in different ways [25], from Chomsky's *cognitive focus* [26], followed by Hymes' *social approximation* [11, 27], to Wiemann's approximation of *relational competence* [28] and the approach based on the *development of language* of Bloom and Lahey [29] and Bryan [30]. Communicative competence in our research is founded on the works carried out by Bachman [7], Canale [31], Hymes [11] and the *Common European Framework of Reference for Language*: *Learning, Teaching, Assessment* (CEFR) [14].

The first allusion to the concept of 'communicative competence' came from Hymes [11]. He defined it as competence that uses the specific knowledge of a language's structure; usually, there is not an awareness of having such knowledge, nor does one spontaneously know how it was acquired. However, the development of communication requires the presence of communicative competence between speakers [25].

Consequently, communicative competence not only is linked with formal aspects imposed from the structure of the language itself (grammatical) but also acquires meaning according to the sociocultural context in which it is developed. The incorporation of these sociocultural communication elements became the pillars of the models developed by Canale [31] and Bachman [7], which is the framework that the CEFR has adopted. In turn, the educational legislation in Spain has also carried out its particular adaptation to the national and regional context with the state regulation [5] and the regional law [32]. The particularity of the adaptation of communicative competence in primary education to national and regional educational legislation has brought about a certain confusion in the Spanish educational panorama. Nevertheless, the diverse conceptualisations of the communicative competence theoretical construct, found in the contributions of Canale [31] and Bachman [7] and in the different Spanish legislations (national and regional), maintain the same basic scheme of communicative competences.

Table 1 shows the correspondences between the different competences of the theoretical proposals of Canale [31], Bachman [7] and the state [33] and regional legislations [34] in Spain. A careful reading of this table highlights how the concept of communicative competence is not affected by the diverse terms used for its designation. Different authors and the legal texts propose the same parameters but present a different degree of specification and depth. The fundamental differences between the theoretical constructs of Canale [31] and Bachman [7] and the state [33] and regional legislations [34] are based on the creation of new competences, such as 'personal competence' (made up of three dimensions—attitude, motivation and individual differences—regarding communicative competence) in the first and 'literary competence' (referring to the reading area, the capacity of enjoying literary texts, etc.) in the second.

**Table 1. Comparative table of the communicative competence components.**

| Canale [31] | Bachman [7] and CEFR | Ordinance 21/01/2015 [33] | BOJA 17/03/2015 [34] |
|---|---|---|---|
| Grammatical (or linguistic) | Organisational-grammatical competence | Linguistic | Linguistic or grammatical |
| Discursive (or pragmatic) | Organisational-textual competence | Pragmatic-discursive | Textual or discursive |
| Sociolinguistic (or sociocultural) | Pragmatic-sociolinguistic competence | Sociocultural | Sociocultural and sociolinguistic |
| Strategic | Strategic competence | Strategic | Semilogical |
| | Pragmatic competence-illocutionary competence | | Strategical or pragmatic |
| | | Personal | |
| | | | Literary |

## 1.3 Communicative competence assessment

Communicative competence assessment must be considered in the process of the communicative teaching-learning of the language ('communicative language teaching' or CLT). This teaching model's axis is 'communicative competence' [35]. This perspective aligns with that of Halliday's systemic functional linguistics [36] and its definitions of the contexts of culture and situation [37]. Savignon's CLT model [38] expands the previous research of Canale and Swain [8] and Canale [31] and adapts communicative competence to a school model (or framework of a competential curriculum). This model develops communicative competence regarding the 'context' and stresses communication's functional character and its interdependence on the context in which it is developed. The communicative competence learning process in primary education is related to the implementation of programmes, which foster the participation of students in a specific communicative context and the regulation of the distinct competences to the social context of the classroom where the learning is performed.

Communicative competence assessment in our research expands upon Lave and Wenger's notion of 'community of practice' [39], the 'theories of genre', which underline the use of language in a specific social context [36, 40], and the 'theory of the socialisation of language' [41, 42]. These notions are integrated into the acts of communication [11, 38, 43] and give rise to diverse communicative competences, which are disaggregated to be assessed.

The changes introduced into the curriculum (with the inclusion of key competences) and in the theories of learning (with the cognitive and constructivist conceptions) have forced the rethinking of assessment [44]. From this perspective, a new evaluation of communicative competence has been constructed from the improvement of the learning processes, not through certain technical measurement requirements [45]. Assessment based on competences or as an investigation has become an excellent model for solving the problem of communicative competence assessment.

Moreover, the modalities of heteroassessment and self-assessment [25] enhance the impact of assessment on children's cognitive development. Basically, there are three factors that influence communicative competence assessment: (a) the culture and context of observation (the culture of the observers is different and makes use of distinct criteria), (b) standards (they cannot be applied to all the individuals of the same community) and (c) conflicts of observation (the valuations of the observations can apply the assessment criteria with a different measurement). Furthermore, Canale and Swain [8] previously underlined the differences between the assessment of the metadiscursive knowledge of competence and the capacity to demonstrate correct use in a real communicative situation. In their reflections, they proposed the need to develop new assessment formats and criteria, which must be centred on communicative skills and their relation between verbal and non-verbal elements.

The perspective adopted in this article sets out from the communicative competence assessment of the analysis of Halliday's systemic functional linguistics [36] and its adaptation to the

School of Sydney's pedagogy of genres developed by Rose and Martin [46]. The School of Sydney's proposal has as its starting point the development of an awareness of the genre in the speaker or writer [47]. Similarly, the discourse's adaptation to the social context at which it is aimed (situation and cultural contexts) has to be taken into account.

In summary, communicative competence assessment sets out from the tools supplied by the analysis of the discourse [48], taking up some elements of diverse discursive traditions, such as pragmatic, conversational analysis and the grammar of discourse (for more information, see [49, 50]). These tools will respond to the levels of the genre, register and language (textual macrostructure and microstructure) [51, 52].

### 1.4 Aims

Setting out from these suppositions, this paper addresses the following aims:

1. To design a communicative competence assessment model based on the Spanish primary education curriculum.

2. To check the effect of the communicative competence assessment model on primary education teachers using a computer application.

## 2 Method

The research design is based on the use of the focus group technique for the study of the same reality, developed through four study groups. Each of these groups represents a school with different characteristics and profiles (see Table 2), enabling a multi-perspective approach, where schools represent different opinions and experiences. The COREQ checklist was followed. All the participants were informed of the nature and aim of the research, thus conforming to the rules of informed consent, and signed written consent forms [dx.doi.org/10.17504/protocols.io.bd8ei9te]. In addition, this research was approved and adhered to the standards of the Social Sciences of the Ethical Committee of Experimentation of the University of Seville.

### 2.1 Participants

Twenty teachers from the second, fourth and sixth years, belonging to four primary education centres in the province of Seville, took part in this study. Prior to consent, participants knew the objectives of the research project and the profiles of the researchers and agreed to collaborate voluntarily in the project. Participants were intentionally selected face-to-face for their diversity in school typology. In this way, participants were obtained from public, private and charter schools. Two of the initially contacted schools refused to participate due to technical problems with their Internet connectivity in the school and the staff's lack of time to attend the training in the evaluation of communicative competence. Participant teachers undertook a training course on communicative competence assessment. The course was developed in the b-learning modality using the Moodle e-learning platform. During the training, teachers

**Table 2. Students taking part in the research.**

| Course | Student experimental group | Student control group | Teachers |
|--------|:--------------------------:|:---------------------:|:--------:|
| 2nd    | 127                        | 91                    | 6        |
| 4th    | 115                        | 96                    | 6        |
| 6th    | 126                        | 98                    | 8        |
| **Totals** | **368**                | **285**               | **20**   |

learned how to use a computer application to assess communicative competence using tablets. This tool, called the 'tool for the assessment of linguistic communication competence' (hereafter, HERACLES), was custom-designed. Later, teachers had the opportunity to implement what had been learned in their classes during a term. The application of the tool took place with 368 students in the experimental group and 285 in the group without the application (see Table 2).

After the application of the tool, the teachers were invited to participate in different discussion groups to note the results of the experience and the effect that HERACLES had on their training. The different focus groups were conducted in teachers' workplaces by the three PhD authors of this paper, one female senior lecturer and two male senior lecturers from the universities of [authors] and experts in educational research. In two of the four schools, members of the management team also attended the focus groups, in addition to participant teachers. The discussion groups were audio-recorded and took place in the educational centres between June and September 2017.

## 2.2 Instruments

The analysis of the audio recordings of the discussion groups and the field notes taken has generated a system of inductive categories (see Table 3). This category system was compiled from the information provided by teachers in the discussion groups. The system of inductive categories was structured through a thematic frame based on the teaching staff's experience in the use of a computer application to assess competence in communication in the classroom. The indicators focused on the ease of use of the computer tool, its usefulness in classroom evaluation, and teachers' assessment of the tool itself. The coding of the discussion group transcripts was performed by the three authors of the current paper. This system has been applied both in the codification phase and in the later analysis of relations with Atlas-ti. The focus group script was designed by the team of authors of this paper and was evaluated by six experts in educational research. Their analysis relied on input based on the understandability of the interview questions and on questions' pertinence to the purpose of the research. The duration of the focus groups was approximately two hours. Recordings' transcriptions were sent to the schools for review. The participants did not make any corrections to the content of the transcripts.

## 2.3 Data analysis

The first aim was accomplished through a comparative analysis of the communicative competence's main components gathered in the models of Canale [31] and Bachman [7] and their

**Table 3. System of inductive categories for communicative competence assessment through a computer application.**

| Categories | Description |
| --- | --- |
| Applicability in daily use | HERACLES enables the optimisation of the assessment |
| Methodological change | The teacher-learning process changes due to the assessment |
| Mistakes of the tablet | HERACLES does not facilitate the teacher-learning process |
| Learning phase | The training of teachers facilitates the implementation of HERACLES |
| Assessment indicators | The indicators of HERACLES facilitate the assessment for learning |
| Insufficient use of functions | Teachers do not use the functions available in HERACLES |
| Positive opinion of the tool | The use of HERACLES facilitates teachers' work |
| Problem in digital literacy | Teachers face difficulty in their digital competence |
| Problems of connections | Teachers have problems related to connectiveness during the use of HERACLES |

relations with both national legislation [33] and regional legislation [32]. This analysis was the basis of the development of a communicative competence assessment model.

The second aim is approached through a qualitative thematic analysis [53, 54] of the discussion groups. The data analysis of the discussion groups' recordings was carried out through Atlas-ti version 6.2. In the operationalisation phase [55], the system of inductive categories [56] was elaborated after listening to all the recordings. The codification of each discussion group was performed a posteriori by three researchers, and the coefficient of agreement between codifiers was calculated via the Fleiss' kappa technique [57, 58].

Fleiss' kappa calculation showed a value of K = 0.91 (see Table 4), which can be described as an excellent interjudge concordance [57]. The disagreement between the different coders was motivated by their interpretation of the application of the transcription categories, which was the result of the inductive process of the creation of the category system. These disagreements were solved through a process of iterative review and clarification of the indicators of the category scheme. After the categorisation of the focal group transcriptions, the three authors of this paper carried out a synthesis and summary of the data. The final report with the results of the research was sent to the different schools for review and feedback.

Finally, we use different analyses of associations and semantic networks [59]. In the search for relations between the codes, we rely on the Atlas-ti Query Tool option. We similarly use the Network tool to carry out the graphic representation of these associations.

## 3 Results

### 3.1 A new communicative competence assessment model

The communicative competence assessment model proposed by Bachman [7] established a clear trend to measure competence as an interpersonal communication product. The elements that it proposes are based on an assessment of both the analysis of the environment of the assessment tasks (environment and type of test) and the indicators that differentiate diverse degrees of achievement of communicative competence in primary education (format, nature of the language, facet of response expected and relation between the input and output information).

The assessment model elaborated (see Table 5) presents the assessment indicators described generally. However, these indicators must be adapted to each of the tasks and genres evaluated in the classroom. The assessment tool was based on the application of distinct elements of the analysis of the discourse and on the selection and transformation of the elements into assessment indicators in the different dimensions. Table 5 presents examples of the assessment indicators related to the following aspects:

a. the levels of discourse (genre, macrostructure and microstructure);

b. the four communicative competence dimensions (speaking, listening, reading and writing); and

**Table 4. Results of Fleiss's kappa.**

| Kappa (K) | ASE | Z-Value | P-Value |
|---|---|---|---|
| .90997038 | .05456660 | 16.67632489 | .00000000*** |

*p < .05, **p < .01, and

***p < .001.

c. the classification of each indicator according to its belonging to various competences (textual, discursive, sociocultural, pragmatic, strategic or semilogical).

The assessment of all these indicators in a school context made the development of the HERACLES computer application for tablets necessary. With this assessment tool (see Figs 1 and 2), it is possible to address not only the broad diversity of assessment indicators but also the heterogeneity of the students themselves, considering their individual variables.

This application enables the carrying out of a learning assessment, providing information concerning the communicative competence teaching-learning process in students during a prolonged period of time. The process assessment can be performed through diverse techniques, such as observation, thinking aloud, or interviews via stimulated recall. Similarly, HERACLES can relate the process' assessment with that of the product through the analysis tools of the oral and written discourse. It was designed to facilitate students' daily follow-up work, streamline the registering of students' communicative competence development, gather information on the teaching-learning process and facilitate decision making for the programming of communicative-competence-related tasks. With this tool, the communicative competence learning assessment process is systematised and allows for the task's assessment to be carried out efficiently and without excessive resource costs in the performance of the teaching work [60].

**Table 5. The communicative competence assessment model.**

| Levels of discourse | | Oral expression | Oral comprehension | Written expression | Written comprehension |
|---|---|---|---|---|---|
| Genre | Macrostructure 1. Global coherence 2. Linear coherence; 3. Textual cohesion | **Textual or discursive** | **Textual or discursive** | **Textual or discursive** | **Textual or discursive** |
| | | a. Structure of the discursive sequence b. Incorporation of narrative, descriptive, expositive and argumentative sequences **c.** Type of discursive interaction | a. Recognises the structure of the discursive sequence b. Extracts the main topic **c.** Identifies the oral genre | a. topStructure of the prototypical textual sequence **b.** Incorporation of the different micropropositions into the dominant macroproposition | a. Identifies the internal organisation of the text (thematic progression) b. Extracts the main or global theme c. Identifies the genre **d.** Recognises the words read previously |
| | | **Sociocultural** | **Sociocultural** | **Sociocultural** | **Sociocultural** |
| | | **a.** Norms of interaction and interpretation (turns speaking, presuppositions, and implications) | **a.** Norms of interaction and interpretation | b. Task of authentic assessment **c.** The discursive genre in the classroom context | **a.** Relates the genre with the communicative aim |
| | | **Pragmatic** | | **Pragmatic** | **Pragmatic** |
| | | **a.** Values the appropriateness of the field, tenor and mode | | a. topAdaptation of the linguistic register to the communicative aim **b.** Appropriateness to the *tenor* | a. Responds to questions of inferential comprehension (not explicit in the text) **b.** Responds to critical questions related to opinion |
| | | **Strategic** | | | |
| | | a. Analysis of non-verbal elements **b.** Analysis of the relationships between non-verbal and verbal elements | | | |
| | Microstructure 1. Lexical level; 2. Syntactic level | **Linguistic** | | | |
| | | a. Scant lexical density and frequent redundancy, catchphrases, code phrases, etc. b. Syntactic complexity and grammatical structures (clauses, groups and phrases) c. Discursive connectors d. Interprets pronouns, pauses and intonations to reinforce the textual cohesion | | | |

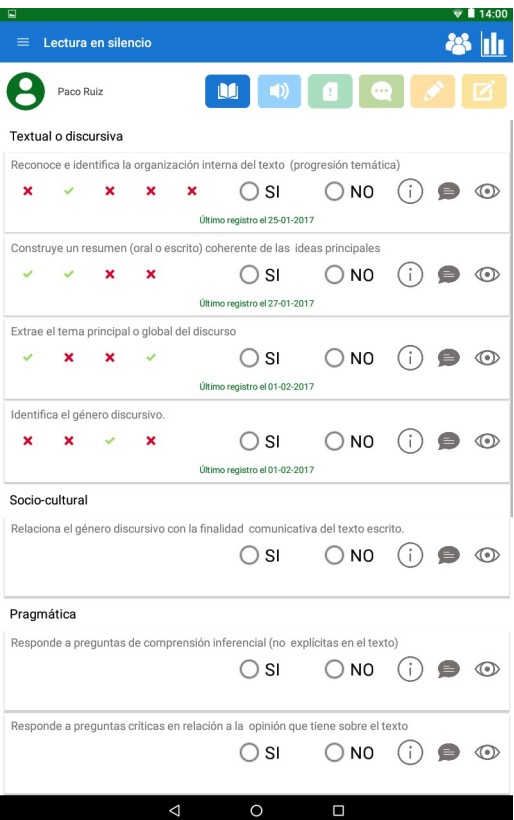

**Fig 1. HERACLES' upper menu.** Reprinted from the COMPLICE project under a CC-BY license.

## 3.2 Effects of the use of the computer application of the communicative competence assessment

The second aim of this research has been addressed from the perspective of the qualitative thematic analysis of the discussion groups. The study of the effect is divided into two perspectives: the positive effects regarding the applicability of HERACLES and teachers' methodological changes and the negative effects of its use. The positive effects have been characterised through causal relations ('cause-effect') or associative relations ('related to') (see Fig 3). The analyses performed have not shown any significant differences between the cases studied. Consequently, in this section, the different cases have not been described separately.

The positive effects are organised into three groups of relations. The first, composed of the causal relation of the applicability of daily use and methodological changes, tackles the changes detected in the methodology when HERACLES has been used with the tablets. In particular, the application of the communicative competence assessment criteria has enabled the improvement of the teaching-learning process in the centres analysed ('the criteria of assessment (. . .) have helped me to focus on teaching' [GD 1]). The communicative competence assessment has led some teachers to modify the assessment process, incorporating feedback ('Yes, there are things I have proposed changing in the assessment: different forms of feedback with the students in the oral expositions and in the reading' [GD 2]) and a process based on the learning assessment and adapted to the context of the classroom ('Everything that is the theme of oral exposition and everything written (summaries) is something that I have had to introduce changes in to spotlight the assessment of the competence' [GD 4]).

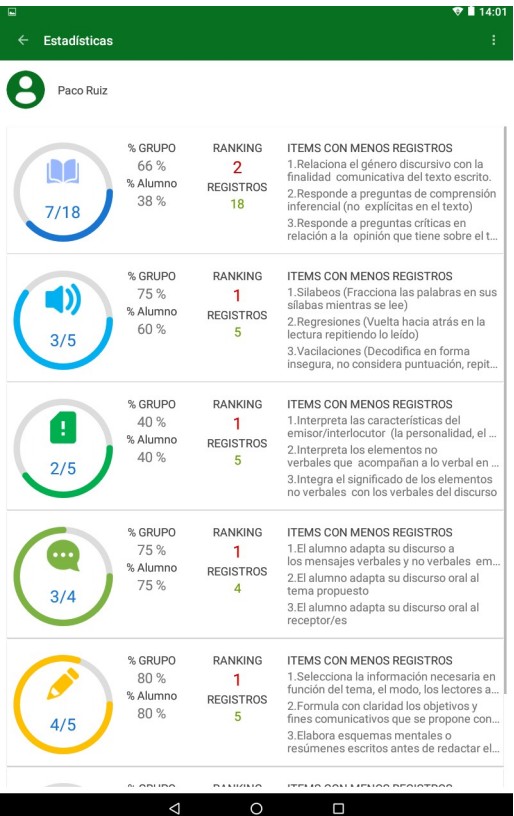

**Fig 2. HERACLES' assessment area.** Reprinted from the COMPLICE project under a CC-BY license.

The second consists of the associative relation between the methodological change and the incorporation of assessment indicators in their daily activity. This has allowed for the evaluation of communicative competence dimensions that were not previously assessed in the classroom ('I have used the tablet (. . .) when the children were speaking: if they gesticulated, if they stared, or if they used the appropriate vocabulary' [GD 1]). In particular, the assessment of oral communication was developed due to the simple use of the tablet as an assessment instrument during the teaching-learning process ('Not a specific activity or day, but rather, it depends on

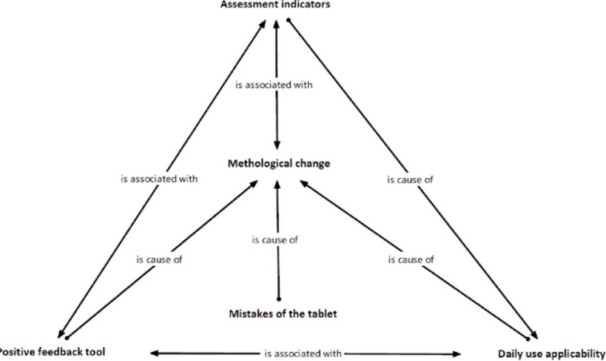

**Fig 3. Graphic representation of the relations between codes about the use of the communicative competence assessment computer application carried out in Atlas-ti.**

the tasks of each subject' [GD 1]. Moreover, the ease of assessing communicative competence in very disparate circumstances within the school day permits this assessment to be extended to different areas of the curriculum ('It was not specifically in the language class but in the classes in which they carried out a task or an activity' [GD 1]). Finally, the use of indicators has generated the teaching perception of a more 'objective' assessment in the classroom ('Assessment is an attitude, and it is very subjective. (. . .) The tool helps me to be more objective' [GD 4]).

A third associative relation is established between maintaining a positive opinion about the use of HERACLES to assess communicative competence and the application of the daily use of the tablet as an assessment instrument. Teachers perceived the use of the assessment with tablets as simple and intuitive ('It seemed to me quite simple and intuitive' [GD 1]). The use of assessment tools and their indicators has led to their use being conceived as something easy and practical for the communicative competence assessment ('It has been much more practical to assess according to the item they asked you' [GD 3]). Similarly, the use of tablets relates HERACLES and its assessment with the facilitators of specific techniques, such as assessment through observation in the classroom ('I would like to use it because it seems handier' [GD 1]), making them quicker and more efficient in the current educational context.

The negative evaluations of the teachers have concentrated on the mistakes of the computer application. The relation between the mistakes and the methodological change is causal. Some difficulties found in the use of HERACLES have led to fewer effects on the methodological change. On the one hand, they are centred on the lack of a button to cancel the different notes recorded ('I would have put the Yes/No option, but I missed the delete option' [GD 1]). On the other hand, the difficulties come from the listing of the students being in an alphabetical order of their first names (and not by their surnames) and of the impossibility of selecting assessment indicators to adapt them to the task assessed and the age of the subjects ('We did not have the option of marking which indicators we wanted to assess and which we did not' [GD 1]). Finally, teachers suggest greater flexibility in being able to incorporate data from the group and students in HERACLES. In this sense, the computer application does not allow for an adaptation to a specific context or the modification of the communicative competence assessment model to adapt it to the programming of the classroom ('I cannot continue using the material because it is closed' [GD 2]).

## 4 Discussion

Our research has addressed the design and effect of a communicative competence assessment model through a computer application. The first aim proposed an evaluation design that facilitates a tool that helps teachers solve the complex process of assessment in the primary education classroom context.

The construction of an assessment model for communicative competence was based on the assessment of the learning concept in the context of the primary education curriculum. This model encourages a deeper analysis of communicative competence, incorporating the different competences involved (linguistic, pragmatic, strategic, etc.). Thus, the assessment of communicative competence (considered a formative assessment) requires a complex process of systematic data collection in the classroom, open to the different indicators determined by the model. In this way, teachers can evaluate communicative competence in different school subjects and develop improvement strategies aimed at one competence or another in a specific and personalised way. This proposal enables a clear heightened awareness of how the discourse has to be assessed, irrespective of the particularities of the assessment activities. This model enables the simple and systematic accessing of the analysis of the oral and written discourse,

making it accessible to both teachers (in summative assessment) and students (through the feedback of the assessment for learning).

The application of this model as an assessment of communicative competence in primary education poses several problems. One of the problems of teachers in communicative competence assessment is the time cost that individualised attention requires. The proposed model advocates for a sustainable assessment [61]. The difficulty of communicative competence assessment requires teachers to address the complexity of the communicative competence teaching-learning process from an individualised perspective. This assessment model allows for reducing the time of this assessment and, in turn, addressing diversity respecting the learning rhythms. The learning assessment will only have an effect in the medium and long term when it is maintained over time. That is, both investment in teachers' training time and handling of the data, which are obtained with computer applications, must be preserved to offer greater rapidity in the feedback and feedforward [18, 62–64].

The second aim presents the effect of the communicative competence assessment model's application on teachers through the use of a custom-designed computer application in primary education. The results reveal a polarisation between two profiles of teachers. The first brings together those who have a positive attitude towards the implementation of new assessment tools. For these teachers, the tool has been useful and has helped improve the communicative competence teaching-learning process. The second model groups those teachers who resist changes to the assessment models. For this group, the implementation of the new model presents numerous difficulties. The motives have resided in the conceptual comprehension of technology in general and of tablets in particular and the resistance to changes in an area such as the culture of school assessment. This resistance to the assessment model's implementation has revealed how primary education assessment processes are the least porous to change in teachers' continuous training process [65].

The assessment model's application has enabled teachers of the first profile to incorporate communicative competence assessment into other curricular areas. The teachers understood that communicative competence assessment must not only be applied to Spanish language and literature. The model's implementation has helped these teachers raise their awareness of assessment for key primary education competences [66].

## 5 Limitations and prospective research directions

The analysis of the research developed in this article has revealed some limitations. The first refers to the communicative competence assessment model. The indicators require teachers to adapt to the different assessment tasks. This possibility must be taken into account in the future development of the HERACLES assessment tool with a view to training the teachers and optimising its use in the classroom.

The effect of the results of the communicative competence model's implementation in the studied centres showed that the processes of change in assessment require a greater time period. In this sense, some of the teachers did not attain a higher degree of advantage and systematicity in the use of the assessment tool model, as individual variables affected this model's rhythm of implementation. Future research projects will have to expand upon the rhythms of learning of the teachers themselves when implementing improvements in the evaluation of the associated key competences.

Relatedly, the use of the HERACLES application presented some difficulties motivated by teachers' scant development of digital competence. Consequently, this has meant a greater investment of time and effort in the adaptation of the assessment model and has brought

about a certain dissatisfaction among participants due to their slow progress in the communicative competence assessment model's changes.

Future works could extend the study to more educational centres that are interested in improving learning assessment. This would give greater potential to the impact it could have on primary education. Similarly, the HERACLES tool must be completed and modified by teachers with the aim of adapting it to each classroom's teaching-learning processes. HERACLES must provide a model that is adapted later by the teacher to systematically and efficiently undertake the communicative competence assessment.

## Supporting information

**S1 Text. Information for teachers in participating school.**
(DOC)

**S1 Data.**
(SAV)

**S2 Data.**
(HPR6)

## Author Contributions

**Conceptualization:** Juan Jesús Torres-Gordillo, Fernando Guzmán-Simón, Beatriz García-Ortiz.

**Data curation:** Juan Jesús Torres-Gordillo, Fernando Guzmán-Simón, Beatriz García-Ortiz.

**Formal analysis:** Juan Jesús Torres-Gordillo, Fernando Guzmán-Simón, Beatriz García-Ortiz.

**Funding acquisition:** Juan Jesús Torres-Gordillo.

**Investigation:** Juan Jesús Torres-Gordillo, Fernando Guzmán-Simón, Beatriz García-Ortiz.

**Methodology:** Juan Jesús Torres-Gordillo.

**Project administration:** Juan Jesús Torres-Gordillo.

**Resources:** Juan Jesús Torres-Gordillo.

**Software:** Juan Jesús Torres-Gordillo.

**Supervision:** Juan Jesús Torres-Gordillo, Fernando Guzmán-Simón, Beatriz García-Ortiz.

**Validation:** Juan Jesús Torres-Gordillo, Fernando Guzmán-Simón.

**Visualization:** Juan Jesús Torres-Gordillo.

**Writing – original draft:** Juan Jesús Torres-Gordillo, Fernando Guzmán-Simón, Beatriz García-Ortiz.

**Writing – review & editing:** Juan Jesús Torres-Gordillo, Fernando Guzmán-Simón, Beatriz García-Ortiz.

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
