## [Decision Letter · Decision Letter 0]

13 Feb 2020

PONE-D-19-33902

Communicative competence in assessment for learning: effect of the application of a model on teachers in Spain

PLOS ONE

Dear Dr. Torres-Gordillo,

Thank you for submitting your manuscript to PLOS ONE. After careful consideration, we feel that it has merit but does not fully meet PLOS ONE’s publication criteria as it currently stands. Therefore, we invite you to submit a revised version of the manuscript that addresses the points raised during the review process.

Athough the topic is interesting, changes are mainly required in the delivery of the content, the introduction and the methods sections, as per reviewers comments.  Given the qualitative nature of the study, the  accuracy of language and content is deemed significant. 

We would appreciate receiving your revised manuscript by Mar 29 2020 11:59PM. To enhance the reproducibility of your results, we recommend that if applicable you deposit your laboratory protocols in protocols.io, where a protocol can be assigned its own identifier (DOI) such that it can be cited independently in the future. For instructions see: http://journals.plos.org/plosone/s/submission-guidelines#loc-laboratory-protocols

We look forward to receiving your revised manuscript.

Kind regards,

Vasileios Stavropoulos

Academic Editor

PLOS ONE

Journal Requirements:

3. PLOS ONE will consider submissions that present new methods, software, or databases as the primary focus of the manuscript if they meet the criteria of utility, validation, and availability described here: http://journals.plos.org/plosone/s/submission-guidelines#loc-methods-software-databases-and-tools. To meet these criteria, please provide supporting materials enabling other teachers and researchers to replicate your teaching intervention such as sample worksheets, a detailed lesson plan or curriculum or other educational materials. If you include supporting materials, they should not be under a copyright more restrictive than CC-BY.

4. Please state specifically whether the IRB approved the study.

5. Please provide additional details regarding participant consent. In the ethics statement in the Methods and online submission information, please ensure that you have specified (1) whether consent was informed and (2) what type you obtained (for instance, written or verbal, and if verbal, how it was documented and witnessed). If your study included minors, state whether you obtained consent from parents or guardians. If the need for consent was waived by the ethics committee, please include this information.

Please provide a copy of the topic guide as Supplementary Information as well as English translations of the screenshots and provided materials.

6. We note that Figures 1 and 2 in your submission contain copyrighted images. All PLOS content is published under the Creative Commons Attribution License (CC BY 4.0), which means that the manuscript, images, and Supporting Information files will be freely available online, and any third party is permitted to access, download, copy, distribute, and use these materials in any way, even commercially, with proper attribution. For more information, see our copyright guidelines: http://journals.plos.org/plosone/s/licenses-and-copyright.

1.         You may seek permission from the original copyright holder of Figures 1 and 2 to publish the content specifically under the CC BY 4.0 license.

Reviewers' comments:

Reviewer's Responses to Questions

**Comments to the Author**

1. Is the manuscript technically sound, and do the data support the conclusions?

Reviewer #1: Yes

Reviewer #2: Partly

2. Has the statistical analysis been performed appropriately and rigorously? 

Reviewer #1: Yes

Reviewer #2: No

3. Have the authors made all data underlying the findings in their manuscript fully available?

Reviewer #1: No

Reviewer #2: Yes

4. Is the manuscript presented in an intelligible fashion and written in standard English?

Reviewer #1: No

Reviewer #2: No

5. Review Comments to the Author

Reviewer #1: The authors of this paper sought to establish a framework for communicative competence (CC) assessment in Spanish classrooms and establish the acceptability of using a tablet-based tool for this assessment. The qualitative methods were appropriate and the authors summarise many useful themes that will be helpful in furthering the design and implementation of their tool and program. However, this manuscript was challenging to read in parts and could benefit from considering the following suggestions for improvement:

* The manuscript requires English editing to improve the clarity. For example the authors state, “This leads is to go deeply into the link…” but it is not clear which linked concepts are being explored. I assume they mean the link between CC in the curriculum and accountability in teaching programs? Phrases such as “Future research projects will have to set out from the rhythms of learning of the teachers themselves when implementing improvements in the evaluation of the key competences” do not quite make sense i.e. it is not clear to an English speaker what “rhythms of learning” means.

* One focus of English editing should be more consistent use of tense. It is not always clear if the authors are describing something that was done or found in their study (past tense) or something that currently occurs in classrooms (present tense).

* The rationale of the study is not clear from the first two paragraphs. The authors clarify this somewhat later, however this rationale should be clear from the beginning of the paper, as should clear definitions of key concepts, particularly CC.

* It is not clear why the learning assessment framework is being introduced where is appears, or what its role is. Is this an established framework for integrating CC into curriculums? Or is this a framework the authors proposed might be useful for achieving this?

* The definition of Learning assessment is circular and should be expanded upon.

* It is not clear to a reader not familiar with this area what the term “Denomination” refers to specifically in this context. Please clarify this.

* I’m not sure what the source “own elaboration” for Table 1 refers to?

* More detail is required regarding how the themes were established and categorised. What established thematic framework was used? Who specifically were the raters? How were disagreements addressed? Who synthesised and summarised the data once it was categorised?

* I am not clear what “ad hoc designed computer application” means. Custom-designed?

Reviewer #2: Overall, this is an interesting and worthwhile area of research, with implications for improving assessment of primary school students. Further review of the paper by the authors is recommended prior to publication, to ensure that all important terms are clearly explained, the rationale for the research is more clearly outlined at the outset, the methodology is clearly linked to the data analysis, and both aims are clearly defined. Please note that the "no" comment to the manuscript being presented in an intelligible fashion is referring to the clarity of writing overall. There are many strong aspects to this paper, but also important areas that could be written more clearly.

Please see below for comments about each section of the paper.

Introduction

The Introduction could be clearer in providing an orientation to the topic and key questions to be addressed. It mentions PIRLS without explaining this, and while this might be common knowledge for those working in education, some brief explanation of PIRLS would be helpful for other readers. Furthermore, it is not quite clear what is being said in regard to improvements or changes in primary school students – when saying, “The results of the improvement in reading of Spanish students in Primary Education (hereafter, PE) present a slight improvement on the PIRLS assessment (2006; 2011; 2016)”, it is not clear what improvements are being discussed, what is meant by a “slight improvement”, and over what time period this is being mentioned (i.e. is it since the last PIRLS assessment?). This slight improvement appears to be discussed as an area of concern, but again, it is unclear why this is of concern, so more background and context here would be helpful.

There is good discussion of what is meant by Communicative Competence; however, it would be helpful to more clearly outline earlier what was meant by the results of CC not being what was expected, so as to clarify how they were not what was expected.

Aims

Generally clear, but Aim 1 could be more specific – given you are discussing PE teachers, it would be good to specify here whether the CC assessment model was based on PE curriculum development.

Method

In this section, please briefly clarify why a multiple case study design has been chosen as the methodological framework, so there is a sound rationale for this.

More detail is needed here to conform to the Consolidated criteria for reporting qualitative studies (COREQ): 32-item checklist, in all domains listed in that checklist. Please review this checklist and add in missing information.

While it appears that participants were teachers rather than students, students are mentioned as if they are part of the sample. It is important to clarify whether they were participants, and if so, what procedures were used for informed consent.

Results

As with the Method, this does not conform to all domains of the COREQ, so further review of these is recommended.

The chosen methodology is a multiple case study design; however, the results do not appear to be written in this manner. You may wish to either reconsider whether the multiple case study approach is appropriate (I would suggest it is not, given it typically involves multiple sources of information for each case), and rewrite this section with a different methodology as appropriate.

Discussion

Generally clear, with the second aim of the study being discussed well. Writing more clearly about the first aim of the study would be helpful here – it is discussed, but could be clearer.

Limitations

Good discussion of limitations, such as potential lack of digital competence within teachers, which is an important area for further education and research.

6. PLOS authors have the option to publish the peer review history of their article (what does this mean?). If published, this will include your full peer review and any attached files.

Reviewer #1: No

Reviewer #2: Yes: Alana Howells

---

## [Author Response · Author response to Decision Letter 0]

25 Mar 2020

Dear reviewers,

1. The authors have incorporated substantial improvements in the text aimed to increase the research rigour of this paper. These modifications have affected all sections of the paper, especially the introduction, methodology and discussion.

2. The authors have not conducted a statistical analysis in this manuscript, as the research design was geared towards the analysis of qualitative data. In the case of Fleiss’ kappa calculation, calculations are correct.

3. The authors have completed the information required by the reviewers throughout the text, and have incorporated new complementary documents related to the development of the research. However, this research must guarantee the anonymity of the participants, as well as the educational institutions to which they belong. These criteria were part of the commitment made by the researchers to the participants before the start of the research.

4. The authors have conducted a thorough review of the academic writing of this paper. To this end, the authors have carefully reviewed the text of the paper and, subsequently, the paper was submitted to the AJE for the review of its academic style. As a result of both revisions, the authors consider that the text is now clearer and more accurate.

Reviewer #1

1. The authors have carried out a profound revision of the academic style of the paper. The modifications to the text have been recommended by AJE and highlighted in the text in grey.

2. The authors have reviewed the use of the paper's verb tenses. The modifications to the text have been recommended by AJE.

3. Rationale of the study:

The authors have modified the rationale of the research and have reviewed the definitions of key concepts (such as communicative competence and assessment for learning). The modifications to the text have been highlighted in the text in grey. The authors have incorporated the following texts into the main paper:

“Assessments carried out by the International Association for the Evaluation of Educational Achievement (IEA) in Spain have provided new evidence for the effects of the educational improvement measures applied in primary education in the last two decades. In particular, the assessment performed in Progress in International Reading Literacy Study (PIRLS) in 2006, 2011 and 2016 has shown how competence in communication in Spanish primary education has not progressed at the rate of that of other European countries [1-3]”.

“Today, we are familiar with numerous definitions of communicative competence [7-13]. The publication in 2001 of the Common European Framework of Reference for Languages [14] has enabled us to describe the skills required for communication and their levels of achievement related to reading, writing, listening and speaking”.

“Consequently, our paper focuses on the design and evaluation of a model for the assessment of communicative competence based on the Spanish curriculum through the use of a custom-designed computer application.”

4. The authors have reviewed the relationship between communicative competence and assessment for learning throughout the paper. The modifications of the text have been highlighted in the text in grey. The authors have incorporated the following texts to the main paper document:

“The design of an assessment model of communicative competence in the school context requires prior reflection related to the dimension assessment model, first, and an assessment of communicative competence, second. Our research started with a reflection on which assessment model for learning was the most appropriate for incorporating communicative competence assessment in the primary education classroom. Assessment for learning is considered an assessment that fosters students’ learning [17-19].”

5. The authors are not drawing on the "learning assessment" concept, but on the “assessment for learning”. The misunderstanding was provoked by a mistake in the translation. In the revision, the authors have applied this later concept throughout the text. 

6. The authors have rephrased the paragraphs indicated by the reviewer. The authors have incorporated the following texts into the main paper:

“Nevertheless, the diverse conceptualisations of communicative competence theoretical construct, found in the contributions of Canale [22] and Bachman [21], and in the different Spanish legislations (national and autonomous), maintain the same basic scheme of communicative competences.”

“A careful reading of this table highlights how the concept of communicative competence is not affected by the diverse names used for its designation. The different authors and the legal texts propose the same parameters but present a different degree of specification and depth.”

7. The authors have removed this comment “own elaboration” for better understanding of the text. 

8. 

a.How the themes were established and categorised. 

The category scheme was developed in an inductive way. That is, the categories emerged directly from the analysis of the information provided by the teachers in the focus groups. The following sentence has been added to the main document: "This category system was compiled from the information provided by teachers in the discussion groups".

b. What established thematic framework was used? 

The authors incorporate the following text into the main document:

"The inductive system of categories was structured through a thematic frame based on the teaching staff’s experience in the use of a computer application to assess competence in communication in the classroom. The indicators focused on the ease of use of the computer tool, its usefulness in classroom evaluation, and teachers' assessment of the tool itself".

c. Who specifically were the raters? 

The authors incorporate the following text into the main document:

" The coding of the discussion group transcripts was performed by the three authors of the current paper".

d. How were disagreements addressed? 

The authors incorporate the following text into the main document:

"The disagreement between the different coders was motivated by their interpretation of the application of the transcription categories, which was the result of the inductive process of the creation of the category system. These disagreements were solved through a process of iterative review and clarification of the indicators of the category scheme".

e. Who synthesised and summarised the data once it was categorised?

The authors incorporate the following text into the main document:

" After the categorisation of the focal group transcriptions, the three authors of this paper carried out a synthesis and summary of the data".

9. The authors have rephrased these paragraphs:

“To check the effect of the communicative competence assessment model on primary education teachers using a computer application.”

“This tool, called the ‘tool for the assessment of linguistic communication competence’ (hereafter, HERACLES), was custom-designed.”

“Consequently, our paper focuses on the design and evaluation of a model for the assessment of communicative competence based on the Spanish curriculum through the use of a custom-designed computer application.”

“The second aim presents the effect of the communicative competence assessment model’s application on teachers through the use of a custom-designed computer application in primary education. The results reveal a polarisation between two profiles of teachers.”

Reviewer #2

1. The authors have incorporated clarifications regarding international PIRLS assessments. The authors have rewritten this section as follows:

“Assessments carried out by the International Association for the Evaluation of Educational Achievement (IEA) in Spain have provided new evidence for the effects of the educational improvement measures applied in primary education in the last two decades. In particular, the assessment performed in Progress in International Reading Literacy Study (PIRLS) in 2006, 2011 and 2016 has shown how competence in communication in Spanish primary education has not progressed at the rate of that of other European countries [1-3]”.

“However, their results have not been expected in the area of communicative competence. Today, we are familiar with numerous definitions of communicative competence [7-13]. The publication in 2001 of the Common European Framework of Reference for Languages [14] has enabled us to describe the skills required for communication and their levels of achievement related to reading, writing, listening and speaking.”

"Consequently, our paper focuses on the design and evaluation of a model for the assessment of communicative competence based on the Spanish curriculum through the use of a custom-designed computer application.”

"The design of an assessment model of communicative competence in the school context requires prior reflection related to the dimension assessment model, first, and an assessment of communicative competence, second. Our research started with a reflection on which assessment model for learning was the most appropriate for incorporating communicative competence assessment in the primary education classroom. Assessment for learning is considered an assessment that fosters students’ learning [17-19]."

2. Aims: The authors have rewritten the aims of this study as follows:

(1) To design a communicative competence assessment model based on the Spanish primary education curriculum.

(2) To check the effect of the communicative competence assessment model on primary education teachers using a computer application.

3. Method:

The authors have reviewed the 32-item checklist and modified the main text when necessary. The following information has been incorporated:

a. The different focus groups were conducted in teachers’ workplaces by the three PhD authors of this paper, one female senior lecturer and two male senior lecturers from the universities of [authors] and experts in educational research.

b. Prior to consent, participants knew the objectives of the research project and the profiles of the researchers and agreed to collaborate voluntarily in the project.

c. Participants were intentionally selected face-to-face for their diversity in school typology. In this way, participants were obtained from public, private and charter schools.

d. Two of the initially contacted schools refused to participate due to technical problems with their Internet connectivity in the school and the staff’s lack of time to attend the training in the evaluation of communicative competence.

e. In two of the four schools, members of the management team also attended the focus groups, in addition to participant teachers.

f. The focus group script was designed by the team of authors of this paper and was evaluated by six experts in educational research. Their analysis relied on input based on the understandability of the interview questions and on questions’ pertinence to the purpose of the research.

g. The analysis of the audio recordings of the discussion groups and the field notes taken has generated a system of inductive categories (see Table 3). 

h. The duration of the focus groups was approximately two hours.

i. Recordings’ transcriptions were sent to the schools for review. The participants did not make any corrections to the content of the transcripts.

j. The final report with the results of the research was sent to the different schools for review and feedback

4. Results:

The authors have revised the methodology to conform to the reviewers' recommendations. The text that clarifies this issue is as follows:

"The research design is based on the use of the focus group technique for the study of the same reality, developed through four study groups. Each of these groups represents a school with different characteristics and profiles (see Table 2), enabling a multi-perspective approach, where schools represent different opinions and experiences. The COREQ checklist was followed. All the participants were informed of the nature and aim of the research, thus conforming to the rules of informed consent, and signed written consent forms".

5. Discussion:

The authors rewrite the Discussion section clarifying those aspects related to the first aim. Thus, the authors incorporate the following fragment to the main text:

“The construction of an assessment model for communicative competence was based on the assessment of the learning concept in the context of the primary education curriculum. This model encourages a deeper analysis of communicative competence, incorporating the different competences involved (linguistic, pragmatic, strategic, etc.). Thus, the assessment of communicative competence (considered a formative assessment) requires a complex process of systematic data collection in the classroom, open to the different indicators determined by the model. In this way, teachers can evaluate communicative competence in different school subjects and develop improvement strategies aimed at one competence or another in a specific and personalised way. This proposal enables a clear heightened awareness of how the discourse has to be assessed, irrespective of the particularities of the assessment activities. This model enables the simple and systematic accessing of the analysis of the oral and written discourse, making it accessible to both teachers (in summative assessment) and students (through the feedback of the assessment for learning)”

“The application of this model as an assessment of communicative competence in primary education poses several problems. One of the problems of teachers in communicative competence assessment is the time cost that individualised attention requires. The proposed model advocates for a sustainable assessment [61]. The difficulty of communicative competence assessment requires teachers to address the complexity of the communicative competence teaching-learning process from an individualised perspective. This assessment model allows for reducing the time of this assessment and, in turn, addressing diversity respecting the learning rhythms. The learning assessment will only have an effect in the medium and long term when it is maintained over time. That is, both investment in teachers’ training time and handling of the data, which are obtained with computer applications, must be preserved to offer greater rapidity in the feedback and feedforward [18, 62-64]”.

6. Limitations:

Thanks.

---

## [Decision Letter · Decision Letter 1]

11 May 2020

Communicative competence assessment for learning: the effect of the application of a model on teachers in Spain

PONE-D-19-33902R1

Dear Dr. Torres-Gordillo,

We are pleased to inform you that based on the two reviewers recommendations your manuscript has been judged scientifically suitable for publication and will be formally accepted for publication once it complies with all outstanding technical requirements.

With kind regards,

Vasileios Stavropoulos

Academic Editor

PLOS ONE

Additional Editor Comments (optional):

Reviewers' comments:

Reviewer's Responses to Questions

**Comments to the Author**

1. If the authors have adequately addressed your comments raised in a previous round of review and you feel that this manuscript is now acceptable for publication, you may indicate that here to bypass the “Comments to the Author” section, enter your conflict of interest statement in the “Confidential to Editor” section, and submit your "Accept" recommendation.

Reviewer #1: All comments have been addressed

Reviewer #2: All comments have been addressed

2. Is the manuscript technically sound, and do the data support the conclusions?

Reviewer #1: Yes

Reviewer #2: Yes

3. Has the statistical analysis been performed appropriately and rigorously? 

Reviewer #1: I Don't Know

Reviewer #2: Yes

4. Have the authors made all data underlying the findings in their manuscript fully available?

Reviewer #1: No

Reviewer #2: Yes

5. Is the manuscript presented in an intelligible fashion and written in standard English?

Reviewer #1: Yes

Reviewer #2: Yes

6. Review Comments to the Author

Reviewer #1: (No Response)

Reviewer #2: Great work on improving this paper. It is now much clearer and straightforward. You have addressed all the issues appropriately.

7. PLOS authors have the option to publish the peer review history of their article (what does this mean?). If published, this will include your full peer review and any attached files.

Reviewer #1: No

Reviewer #2: Yes: Alana Howells

---

## [Editor Report · Acceptance letter]

20 May 2020

PONE-D-19-33902R1 

Communicative competence assessment for learning: the effect of the application of a model on teachers in Spain 

Dear Dr. Torres-Gordillo:

I am pleased to inform you that your manuscript has been deemed suitable for publication in PLOS ONE. Congratulations! Your manuscript is now with our production department. 

With kind regards,

on behalf of

Dr. Vasileios Stavropoulos 

Academic Editor

PLOS ONE